# An Electrochemical Tilt Sensor with Double-Band Electrodes Fabricated by Wire Winding

**DOI:** 10.3390/mi13081229

**Published:** 2022-07-31

**Authors:** Yik-Kin Cheung, Hongyu Yu

**Affiliations:** 1Department of Mechanical and Aerospace Engineering, The Hong Kong University of Science and Technology, Hong Kong, China; ykcheungab@connect.ust.hk; 2HKUST Shenzhen-Hong Kong Collaborative Innovation Research Institute, Shenzhen 518048, China

**Keywords:** tilt sensor, inclinometer, molecular electronic transducer, electrochemical sensor, amperometry sensor, 3D printing, double-band electrode

## Abstract

This paper presents the principle, design, fabrication, and characterization of Molecular Electronic Transducer (MET) dual-axis tilt sensors. The proposed sensor has a 3D-printed cylindrical channel inserted with four double-band electrodes and partially filled with a liquid electrolyte. The double-band electrodes were fabricated by wire winding with a ~0.1 mm anode-cathode distance under controlled tension. It allows the electrode to become any 3D coil rather than a 2D structure by microfabrication and exhibits good repeatability (±10%). The tilting changes the electrolyte level and electrode-electrolyte contact area, resulting in Faradaic current changes. The *x*-axis and the *y*-axis sensitivity reach 0.121 V/° and 0.154 V/°, respectively.

## 1. Introduction

The digital tilt sensors measure the absolute inclination of the Earth’s gravitational direction. This unique ability is indispensable in applications, including apparatus leveling, solar tracking, structural monitoring, automotive safety, navigation, etc. There are various sensing mechanisms with either a solid or liquid pendulum. The solid pendulum has a better dynamic response and is minaturizable. In contrast, the liquid pendulum suits high-resolution static tilt sensing but requires a large housing to overcome the capillary forces.

The common commercialized digital tilt sensors are tri-axis microelectromechanical systems (MEMS) accelerometers and liquid electrolytic tilt sensors. The former measures acceleration in three orthogonal directions and calculates the gravity direction by trigonometric operation. The latter has a container partially filled with conductive liquid and the immersing electrodes. The electrical resistance between the electrodes changes during tilting, and a bridge signal conditioning circuit can extract the tilt angle. However, an AC excitation voltage is required to prevent electrolysis, resulting in complex and costly circuitry. 

Various tilt sensors with different mechanisms are under research, including optical, capacitive, electric contact (tilt switch), thermal, and magnetic. 

The optical inclinometers received attention due to their immunity to electromagnetic interference (EMI), which suits harsh environments and provides potentially high performance. The majority are the Fiber Bragg Grating (FBG) sensors [1,2,3,4,5,6,7,8,9,10,11,12] that reflect a particular light frequency according to the optical fiber strain. Other methods include the Fabry–Perot [13,14] optical resonator, bubble-level with photodiodes [15], and transmission grating [16]. Their resolution is high, down to 0.000383° [13] by the Fabry–Perot method, and the FBG can achieve 0.0024° [7] but requires a complex setup. 

Capacitive sensing is mature and widely employed in the MEMS sensor. On the other hand, researchers have developed dielectric liquid capacitive tilt sensors. However, the slow liquid motion contradicts its fast response advantage. Further, the best-reported resolution is only 0.1° [17], comparable to most solid-state MEMS tilt sensors due to their susceptibility to EMI and environmental conditions, such as humidity.

The electric contact tilt sensors extend the simple tilt switch by introducing multiple contacts [18,19,20,21]. However, traditionally filled mercury is volatile and toxic, giving rise to safety concerns. Novel materials can improve safety, such as carbon nanotube dispersion in water [22], solid solder sphere [21], and Gallium-based alloy. However, they only output discrete angles and thus are low resolution.

The thermal tilt sensor has microheaters to generate natural convection and thermistors to detect gravity direction [22,23,24,25,26,27]. The authors of ref. [26] reported a MEMS thermal tilt sensor with a resolution of 0.003° and a response time of 0.6 s with SF6 gas medium, but with a high power consumption of 45 mW. The carbon-based sensors [25,27] have a lower power consumption with good sensitivity. 

Finally, the magnetic tilt sensor [28,29,30] uses a ferrofluid and achieves a resolution of 0.004° [30]. 

Previous studies developed electrochemical tilt sensors based on the molecular electronic transducer (MET) principle [31,32]. The amperometry sensor measures the current proportional to the electrode-electrolyte contact area and its change when tilted. It requires low voltage (<1 V) DC excitation and has a high sensitivity, low power consumption, and low susceptibility to EMI due to the current signal. This paper presents a dual-axis MET tilt sensor fabricated by a simple wire-winding process and 3D printing, in contrast to the previous microfabricated single-axis one. Wire winding enables the fabrication of 3D electrodes, compared to 2D patterns by microfabrication.

### Sensing Principle

Figure 1a illustrates that the MET tilt sensor has a cylindrical housing partially filled with the electrolyte (2 M KI + 50 mM I_2_) and four double-band electrodes located at the radius midpoints with a 90° separation. A voltage (0.7 V) across the anode and cathode drives the reversible electrolytic reaction:$$3I^{-}⇋I_{3}^{-}+2e^{-}$$

Its current is described by:I=Dq∮(∇c,n)dS
where I is current; D is the diffusion coefficient; ∇c is the concentration gradient; n is the electrode surface normal vector; dS is the electrode surface area element. 

In Figure 1b, the immersion depths for the double-band electrodes 1–4 are L_1_ to L_4_, respectively. Electrodes 1 and 2 form the *x*-axis, while electrodes 3 and 4 form the *y*-axis tilt sensor. When tilted along the *x*-axis, L_2_ becomes longer than L_1_, the same as their reaction current. On the other hand, the L_3_ and L_4_ are equal and the same as the zero tilt situation.

Figure 1c presents a geometrical analysis of the current-angle dependency:dI∝dA∝dl=xtanθ

The current from electrodes 1 and 2 increases and decreases by dI, respectively, which is proportional to the electrolyte–electrode surface area changes dA, and the immersion depth changes dl. The immersion depth changes can be derived from the trigonometric relationships, and the sensor output is proportional to the tangent of the tilt angle, tanθ, which can be considered as linear if $\left|\theta \right|<20^{{}^{\circ}}$. Figure 1d illustrates simultaneous tilting on two axes. L_2_ is longer than L_1_ along the *x*-axis, and L_4_ is longer than L_3_ along the *y*-axis. Note that L_1_ and L_2_ in Figure 1d are the same as in Figure 1b, as their *x*-axis tilt angles are the same.

## 2. Materials and Methods

### 2.1. Fabrication

Figure 2 shows the fabrication processes of the proposed sensor divided into two parts: double-band electrode fabrication and assembly. 

The double-band electrode comprises three parallel 0.1 mm diameter wires in contact (platinum wire, insulated wire, platinum wire) by winding the rectangular coils on a laser-cut steel plate. First, pass the platinum wire through the hole in the steel plate and stick tape to fix the open end. Another end of the wires is held on a spool located on a wire releaser with a tunable tension, as shown in Figure 3d. Then, rotate the steel plate to wind the Pt wire for several turns under a tension of ~2.3 N for straightening. Cut the wire after winding and fix the open end with another tape. Repeat the processes for an enamel insulated copper wire under a tension of ~1 N and platinum wire under ~2.3N, and ensure they are in contact with the previous coil. After completing the winding of the three wires, it forms the double-band electrodes with Pt/insulated/Pt wires. Then, clean the surface of the wires with IPA and air dry before sticking a Kapton tape to fix their relative position. Cut the wires near the edge of the Kapton tape and transfer the released assembly onto a glass slide. Finally, trim and separate the individual double-band electrodes, following the edge of the Pt wires, with a surgical blade. 

For the assembly, the sensor consists of a 3D-printed rigid housing (Clear resin, Form 3, Formlab, Somerville, MA, USA) and 3D-printed soft plugs (Flexible 80A resin, Form 3, Formlab, USA) for inserting the prepared double-band electrodes. The soft plugs enable an electrode replacement in case of failure, such as the detachment of the Pt wires from the Kapton tape short circuit. First, coat the housing with 5 μm thick Parylene C, using PDS2010 Labcoter (Specialty Coating Systems, Indianapolis, IN, USA) as a passivation layer and Fluropel 1601 hydrophobic coating (Cyontix, Beltsville, MD, USA). Then insert a top soft plug and electrode into the plug. Next, insert the bottom plug with its insertion hole aligned to the electrode. Seal the top and bottom plugs with UV glue and solder the Pt wires to the external electrical connections. Repeat the processes for all of the four electrodes. Finally, inject 0.8 mL of the electrolyte composed of 2 M KI + 50 mM I_2_ in a water-ethanol mixture (1:2 ratio) with a syringe and needle through a hole on the top.

### 2.2. Setup

Figure 3a shows the tilt sensor testing setup. The tilt sensor, circuit, and reference sensor are mounted on a single-axis turntable controlled by Arduino. The output from the reference (PWM) and the test sensor (voltage) are recorded by a data-acquisition device (USB-6346, National Instruments, USA). The pneumatic vibration isolation table prevents ambient mechanical noise. The inset figure (b) is a zoom-in view of the testing payload.

Figure 3c shows the signal conditioning circuit schematic for one axis and the corresponding component values for the two axes. Each double-band electrode contains an anode and a cathode. A 0.7 V bias voltage across the anode and cathode drives the reaction in the mass-transport limited condition without electrolysis, with the cathodes as the output terminals. The cathode output current converts to a voltage signal by a transimpedance amplifier with a lowpass filter cutoff at 50 Hz. The voltage output from the two transimpedance amplifiers is fed into the instrumentation amplifier (gain of 39) to obtain a single voltage. In the Figure 3c inset table, the component values for the *x*-axis transimpedance amplifiers are equal, as the two double-band electrodes exhibit an acceptable uniformity. Meanwhile, for the *y*-axis, two sets of component values compensate for the electrode asymmetry. Note that it will not affect the linearity of the sensor. The output for the *y*-axis is:$$V_{out}=I_{2,0}R_{2}+C_{2}R_{2}x\tan\theta -I_{1,0}R_{1}-C_{1}R_{1}x\tan\theta \;=\left(I_{2,0}R_{2}-I_{1,0}R_{1}\right)+x\tan\theta \left(C_{2}R_{2}-C_{1}R_{1}\right)$$
where $I_{2,0}$ and $I_{1,0}$ are the zero tilt angle currents; and $C_{1},\;C_{2}$ are the proportional constant between the current and tilt angle of electrodes 1 and 2, respectively. The derived result shows that the output voltage remains directly proportional to tanθ.

## 3. Results

This section first presents the chronoamperometry (CA), cyclic voltammetry (CV), and electrochemical impedance spectroscopy (EIS) of the four platinum double-band electrodes.

Then, the assembled tilt sensor tilts from −25° to +25 ° and to −25°, with a 5° step along the two sensing axes. The voltage against the tilt angle and the sensor output during experiments are shown and discussed.

### 3.1. Electrochemical Characterization

Figure 4a shows the cyclic voltammetry (CV), and Figure 4b shows the chronoamperometry (CA) of the four double-band electrodes with a two-electrode system. The anodic/cathodic peaks are the oxidation/reduction of the iodide/triiodide redox couple [33] and are symmetrical because of reversibility. The CV and CA results are similar for all of the electrodes, except for the right pair in the positive voltage region of CV. The variations could be attributed to the fabrication error. Figure 4c shows the zoom-in of the CA near 100 s; the currents after 100 s that are within ±10% indicate that the winded electrodes have a good repeatability.

Figure 4d shows the EIS with an amplitude of 10 mV, centered at the open circuit potential from 0.05 Hz to 100 kHz. Two semicircles are shown and overlapped in the high and intermediate frequency regions. The first could be the double-layer capacitance and the charge transfer resistance as usual. On the other hand, the second one may be due to material adhered to the electrode surface (i.e., the Kapton tape) [34]. Table 1 shows the fitted element values using the equivalent circuit shown in Figure 4d (inset).

### 3.2. Sensor Performance

Figure 5a,b shows the voltage against the tilt angle from −25° to +25° with a 5° step angle for the *x*-axis and *y*-axis, respectively. The forward curve sweeps from −25° to +25° and the backward curve is the opposite. From the linear regression, the sensitivity of the *x*-axis is 0.121 V/° and 0.154 V/° for the *y*-axis. The CA above shows that the *x*-axis steady-state currents are higher than the *y*-axis. Therefore, their sensitivity difference is due to the higher resistor value for electrode asymmetry compensation (Section 2.2) of the *y*-axis. The asymmetry generates a large zero tilt voltage on the *y*-axis (2.124 V) but small for the *x*-axis (−0.6682 V). 

Figure 5c,d plots the output waveform during the *x*-axis and *y*-axis tilting experiments. Around half of the stepping could result in a flat output, indicating that the electrolyte surface tension and viscosity are suited for the tilt sensor application. The non-steady waveform could be due to the contact angle hysteresis from the rough 3D-printed surface and the double-band electrode. Figure 6a,b shows the photographs with the ideal and imperfect electrolyte meniscus on the inner wall of the 3D-printed housing.

The spikes in the non-sensing axis data are proportional to the cross-sensitivity. The cross-sensitivity of the *y*-axis in Figure 5c is higher than that of the *x*-axis in Figure 5d. Furthermore, from Figure 5c, the cross-sensitivity is largest between +5° to +20°. In contrast, from Figure 5d, the cross-sensitivity is largest at +20°. Therefore, apart from the electrode misalignment from the centerline, another contribution could be electrode-curved at a particular position, leading to a tilt-dependent cross-sensitivity.

The previous study shows that there are always transient spikes at the beginning of the tilting [31], originating from the moving electrolyte–electrode contact. They are missing in the stepping between −10° between 0° in Figure 5c, but a voltage step remains. The flattening may be due to capillary forces slowing electrolyte movement on the electrode surface.

The previous study shows that the most considerable hysteresis happens when reversing the tilting direction due to contact angle hysteresis [31]. However, the most prominent hysteresis occurs at +5° in Figure 5a, and +15° in Figure 5b. It is because the electrode is no longer at the side wall, and thus, the effect of the contact angle hysteresis becomes smaller. On the other hand, the meniscus imperfection (Figure 6b) influences the liquid level and generates unpredictable hysteresis.

### 3.3. Discussion

The sensor has a low power consumption, a simple circuit, and high sensitivity. However, there are a few limitations. 

First, the sensor has a long startup time of more than 20 s from Figure 4b due to the diffusion of the ions to a quasi-steady state. Similarly, the response time is a few seconds when tilted at 5°. Secondly, the tilting changes the electrode-electrolyte contact area and generates large transient spikes. They could trigger a false alarm if a thresholding safety mechanism is required. In addition, it indicates high sensitivity to the external dynamic vibrations and a significant error in the dynamic conditions. Thirdly, a large hysteresis due to 3D printing can be solved by alternative manufacturing processes and materials. Finally, the sensitivity increases with the temperature due to a faster diffusion rate. Therefore, the sensor is suitable for indoor static sensing requiring low-power consumption.

## 4. Conclusions

In conclusion, this paper presents a molecular electronic transducer (MET) tilt sensor containing a 3D cylindrical housing and four double-band electrodes fabricated by wire winding. The sensor is suitable for indoor static sensing requiring low power consumption. It allows the double-band electrodes to form any 3D coil instead of a 2D structure by microfabrication. The electrochemical data show good repeatability (±10%) among fabricated electrodes. The tilt sensor has a sensitivity of 0.121 V/° and 0.154 V/° on the *x*-axis and the *y*-axis, respectively. However, unpredictable hysteresis is significant due to the rough 3D-printed housing resisting the electrolyte movement.

## Figures and Tables

**Figure 1 micromachines-13-01229-f001:**
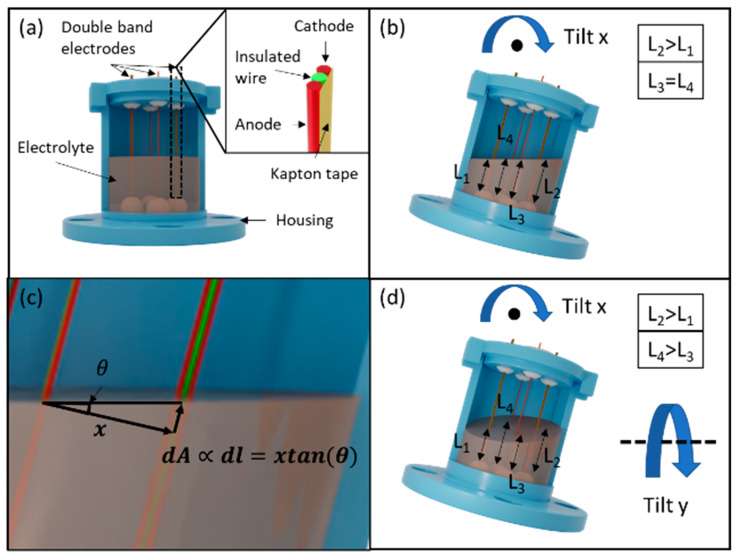
(**a**) The structure of the dual-axis MET tilt sensor; (**b**) the illustration of the sensor tilted along the *x*-axis; (**c**) zoom-in view and trigonometry analysis of the *x*-axis tilt; (**d**) the illustration of simultaneous tilting in both the *x*-axis and *y*-axis.

**Figure 2 micromachines-13-01229-f002:**
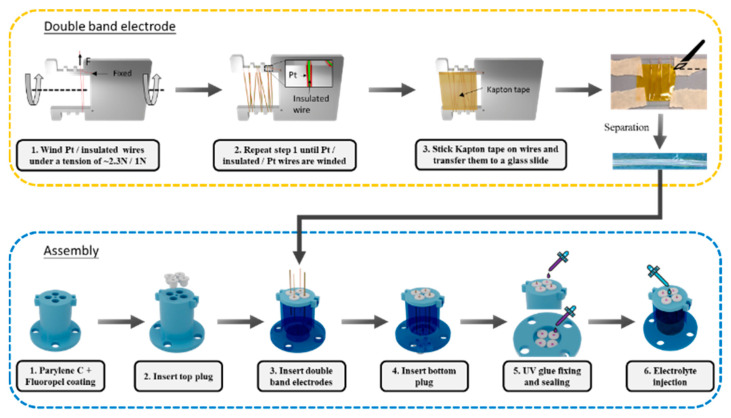
The fabrication of double-band electrodes (**top**) and the tilt sensor assembly process (**bottom**).

**Figure 3 micromachines-13-01229-f003:**
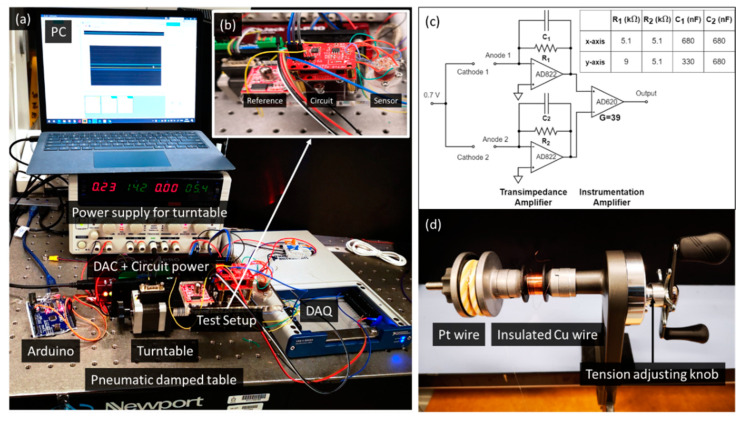
(**a**) The experimental setup of the tilt sensor; (**b**) test setup including a reference sensor, circuit, and the tilt sensor are mounted on a turntable; (**c**) readout circuit and components value for the *x*-axis and the *y*-axis; (**d**) a photograph of the wire releaser that has controllable release tension with the Pt wire and insulated Cu wire spool.

**Figure 4 micromachines-13-01229-f004:**
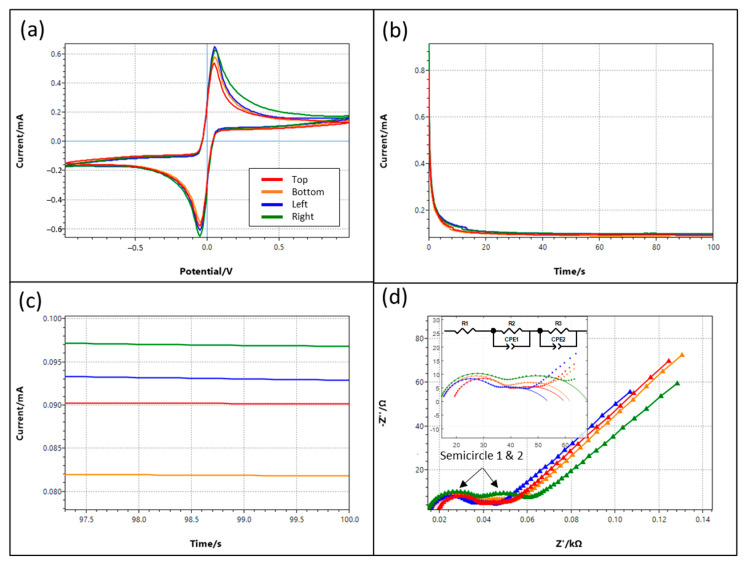
Electrochemical characterization of the four double-band electrodes, top and bottom, is the *y*-axis while left and right are the *x*-axis (**a**) Cyclic voltammetry, the anodic peak is the oxidation of iodide to triiodide ions, and the cathodic peak is the reduction of triiodide ions to iodide ions; (**b**) Chronoamperometry (CA); (**c**) zoom-in view of CA in (**b**); (**d**) electrochemical impedance spectroscopy (EIS), inset figure shows the EIS (dots) and equivalent circuit fitting (lines) within the high and intermediate frequency range. The equivalent circuit consists of three series elements: one resistor and two blocks with a resistor parallel to a constant phase element (CPE).

**Figure 5 micromachines-13-01229-f005:**
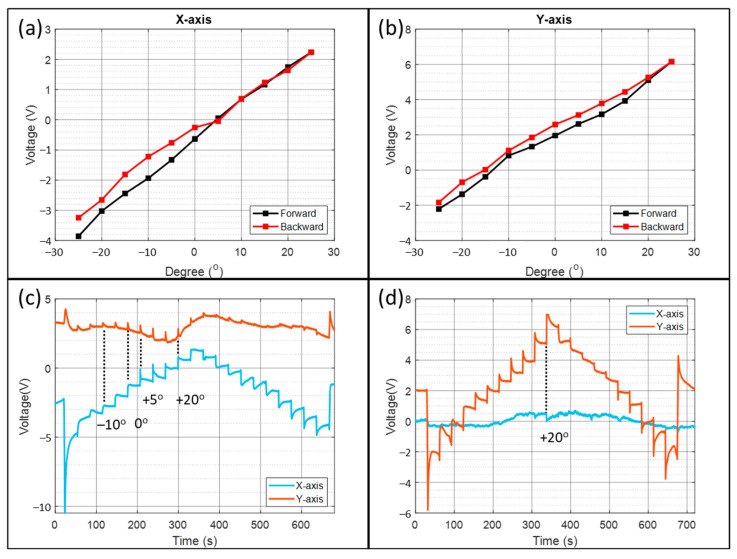
Performance of the tilt sensor: (**a**) *X*-axis voltage against tilt angle; (**b**) *Y*-axis voltage against tilt angle; (**c**) voltage against time when tilting along the *x*-axis; (**d**) voltage against time when tilting along the *y*-axis.

**Figure 6 micromachines-13-01229-f006:**
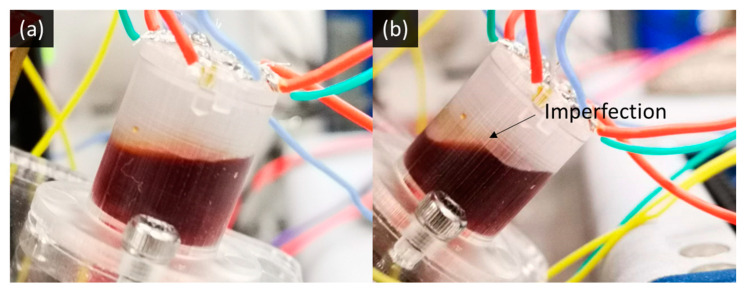
(**a**) The electrolyte meniscus in ideal condition; (**b**) the imperfect electrolyte meniscus.

**Table 1 micromachines-13-01229-t001:** Equivalent circuit fitting results of EIS.

Element	Top	Bottom	Left	Right
R1 (Ω)	18.75	15.07	14.63	14.01
R2 (Ω)	17.5	20.6	15.6	20.9
|CPE1| (T)	1.6458 × 10^−6^	2.0612 × 10^−6^	1.5877 × 10^−6^	2.571 × 10^−6^
∠CPE1 (ϕ)	0.89479	0.86299	0.91045	0.86933
R3 (Ω)	23.34	26.09	23.38	33.36
|CPE2| (T)	0.0011689	0.0006357	0.0011073	0.00038612
∠CPE2 (ϕ)	0.53648	0.60159	0.50492	0.62528

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
