# Peer review of "An Electrochemical Tilt Sensor with Double-Band Electrodes Fabricated by Wire Winding"

_micromachines, 2022, doi:10.3390/mi13081229_

Round 1

Reviewer 1 Report

The manuscript describes the development of tilt sensors based on electrochemical measurements. The work is interesting but I would like the authors to better justify the electrochemical characterization. Examples:

- Fig. 4A What are the redox processes related to anodic and cathodic peaks.

- Fig. 4D ISE The authors describe that: "there are two semicircles shown and overlapped in 1the low-frequency region. It could indicate effects from simultaneous ion transport in the electrolyte and the ion intercalation into the electrode. The first semicircle corresponds to the bulk electrolyte resistance, and the second one corresponds to charge transfer / mass transfer resistance"

I think the authors have confused this interpretation. The two semicircles are set for high frequency to medium frequency. At low frequency it corresponds to mass transport. As we can see in the figure the presence of a warburg component. Just check the 45 degree angle. The first capacitive arc corresponds to the double layer capacitance and charge transfer. Solution resistance is checked at the start of the arc. I believe that the second semicircle is characteristic of the material adsorbed on the surface of the electrodes.

It would be interesting for the authors to add the equivalent circle obtained by the impedance spectra.

Reviewer 2 Report

The presented work concerns the manufacture, construction and description of the principle of operation of the electric contact tilt sensor. 

A prototype of the device was made and then and research on its usefulness for measuring the angle of tilt with respect to two axes.  In my opinion, the work is interesting, and the proposed solution can find numerous practical applications. I found no substantive errors in the work. However, the authors should specify the limitations of the proposed design solution related to, for example:

-speed of response of the sensor

- dynamic error of the sensor

Moreover, the authors should specify how the measurement will be affected by vibrations of the device in which the sensor will potentially be used.

Round 2

Reviewer 1 Report

The authors answered all questions raised by the reviewer. As well, they made the necessary changes to the manuscript.